# Advances in Therapy of Adult Patients with Acute Lymphoblastic Leukemia

**DOI:** 10.3390/cells14050371

**Published:** 2025-03-04

**Authors:** Oscar Sucre, Saagar Pamulapati, Zeeshan Muzammil, Jacob Bitran

**Affiliations:** 1Department of Hematology and Medical Oncology, Advocate Lutheran General Hospital, Park Ridge, IL 60068, USA; oscar.sucre@aah.org (O.S.); saagar.pamulapati@aah.org (S.P.); 2Chicago Medical School, Rosalind Franklin University of Medicine and Science, North Chicago, IL 60064, USA; zeeshan.muzammil@rosalindfranklin.edu

**Keywords:** lymphoblastic, leukemia, treatment, novel, targeted, bispecific, ADC

## Abstract

The landscape of adult acute lymphoblastic leukemia (ALL) is dramatically changing. With very promising results seen with novel immunotherapeutics in the setting of relapsed and refractory disease, the prospect of using these agents in first-line therapy has prompted the development of multiple clinical trials addressing this question. This review seeks to outline and expand the current standard of care, as well as new advances, in the treatment of adult patients with ALL and address future areas of research. We expect the frontline integration of immuno-oncology agents such as bispecific T-cell engagers, antibody–drug conjugates, and chimeric antigen receptor (CAR) T cells may maintain or improve outcomes in adults while also minimizing toxicity. Treatment of ALL will continue to evolve as we focus on personalized, patient-centered approaches.

## 1. Introduction

Acute lymphoblastic leukemia (ALL) is a hematologic malignancy caused by dysregulated proliferation of immature lymphoid cells. This dysregulation primarily occurs in the bone marrow but can extend extramedullary to the peripheral blood or various organ systems. The uncontrolled differentiation and spread of leukemia is engendered by alterations at the chromosomal and genetic level. These genetic mutations ultimately guide the classification, sub-classification, prognosis, and treatment of this complex malignancy [1,2].

While ALL is primarily a pediatric leukemia, it can also occur in adults, accounting for approximately 20% of leukemias that occur in the adult population. Given its predominance in the pediatric population, changes and improvements in the treatment of ALL typically are first demonstrated in children and then extended and further studied in adults. Of note, cure rates in children with ALL have exceeded 90% in recent years [3,4].

Unfortunately, despite the use of pediatric-based regimens and novel immunotherapeutics or other targeted therapies, cure rates in adults with ALL remain low, with a 5-year overall survival of only 20–40% of patients. For patients older than 70 years of age with ALL, the long-term survival is even lower at less than 5% [5].

Therefore, continued research into how to improve ALL treatment remains vital in the effort to improve remission rates, overall survival, and treatment-related toxicities. Current guidelines for the treatment of adult ALL typically involve an algorithmic approach utilizing multi-agent chemotherapy regimens and the consideration of allogeneic hematopoietic stem cell transplantation (HCT) for patients with a high risk of relapsed disease. Regimens are further subdivided into specific phases of induction, consolidation, and maintenance therapy. Measurable residual disease (MRD) testing has also become an essential part in guiding the choice and intensity of these various phases [2].

Recently, genetic profiling has also become paramount with the use of targeted therapies. For example, in *Philadelphia chromosome positive (Ph+)* ALL, the use of tyrosine kinase inhibitors (TKIs) has significantly improved outcomes when combined with chemotherapy [6]. As is the case for many other malignancies, immunotherapy has also been extensively studied in ALL and has shown early promise, with the incorporation of monoclonal antibodies, such as rituximab, inotuzumab ozogamicin, and the bispecific T-cell engager blinatumomab, into treatment regimens [7]. In addition, ongoing preclinical studies and clinical trials underscore the potential of B-cell lymphoma 2 (BCL-2) homology 3 (BH3) mimetics as a novel therapeutic approach, particularly for patients with relapsed or refractory disease [8]. This paper seeks to outline current approaches and guidelines to the classification and subsequent treatment of ALL in adults, synthesizing findings from recent studies to examine the efficacy of current treatment modalities. By doing so, we hope to highlight advances and ongoing challenges, with a specific focus on emerging therapies and treatment paradigms that have shown promise in the refractory or relapsed setting and may soon be introduced as frontline therapy.

## 2. Fit Older Adults

In medically fit older adults with ALL, treatment plans are highly individualized, as there is no standard chemotherapy regimen. Fitness assessments typically use validated scoring systems such as the Charlson Comorbidity Index (CCI) and the Hematopoietic Cell Transplantation-Specific Comorbidity Index (HCT-CI, also known as Sorror Score), which help stratify patients based on their ability to tolerate intensive therapy. Generally, patients aged 60–75 years with a low comorbidity burden (CCI ≤ 2, HCT-CI ≤ 3) are considered fit for modified intensive regimens, whereas those with multiple comorbidities or aged over 75 years may be classified as unfit.

Initial management often includes leukoreduction with steroids, with the possible addition of cyclophosphamide or vincristine. Anthracyclines are typically avoided in this population due to their association with increased mortality. Asparaginase remains a key agent in ALL therapy. Its role in pediatric-inspired regimens for young adults has improved outcomes, but toxicity concerns limit its use in older patients. Pegylated formulations and novel enzyme depletion strategies may allow broader application in adult populations [9]. In the United States, a modified regimen known as mini-HCVD (consisting of dose-reduced cyclophosphamide, dexamethasone, cytarabine and methotrexate) is commonly used for B-cell ALL. In Europe, a mini version of the pediatric Berlin–Frankfurt–Munster (BFM)-based chemotherapy is employed. Both approaches have demonstrated complete remission (CR) rates of 70–80%, but they also present significant drawbacks. The mortality rate during induction therapy ranges from 10 to 20%, with a higher risk for those aged over 75 years, where it can approach 40%. Even among those achieving CR, the overall mortality rate remains around 30–40% [10].

The conclusion from these approaches is that chemotherapy serves best as a backbone to more targeted therapies. While rituximab shows benefits in younger adults, it is less effective in older adults due to increased risks of infection complications. Inotuzumab ozogamicin, an antibody–drug conjugate (ADC), first showed efficacy against relapsed/refractory B-cell ALL in adults in the phase III INO-VATE trial. This led to increasing interest in incorporating this medication to the frontline setting [11].

In the first-line setting, early data from a study combining mini-HCVD with inotuzumab ozogamicin followed by three years of 6-mercaptopurine, vincristine, methotrexate and prednisone (POMP) consolidation (compared to chemotherapy alone) in adults over 60 years old suggested favorable outcomes. The relapse-free survival (RFS) at three years was 49%, and overall survival (OS) was 56%, which compared favorably to historical data. Although high mortality during CR induction (about 25%) was observed, 85% of patients achieved CR. However, there was a high incidence of veno-occlusive disease (VOD) from inotuzumab ozogamicin, which led to dose reductions in this drug [12]. One study in 2022 with older adults with philadelphia negative (Ph−) ALL receiving inotuzumab ozogamicin, mini-HCVD, and/or blinatumomab consolidation (a bispecific CD19 and CD3 antibody) showed even higher CR rates (98% vs. 88%), with fewer early deaths (0–8%), and a decrease in deaths in patients with CR (5–17%). Despite this, significant toxicity prompted protocol amendments, including reducing induction cycles from four to two and replacing POMP with blinatumomab consolidation for adults over 70 years old. The results showed superior RFS at 64% versus 34%, and OS at 63% versus 34%. However, it is important to note that this approach is still not approved for first-line use outside of clinical trials [11,13].

Recent updates on inotuzumab ozogamicin in first-line therapy show promising results. A phase II study combining HyperCVAD with blinatumomab and POMP consolidation, with the addition of inotuzumab ozogamicin during induction, showed excellent outcomes in patients under 60 years. At the three-year follow-up, 100% of patients achieved CR, and 91% were minimal residual disease (MRD)-negative after one cycle. The OS was 84%, with only 10% of patients relapsing. The treatment was well tolerated, with no evidence of VOD. The safety profile was generally favorable, with only one patient discontinuing blinatumomab due to grade 2 encephalopathy [11,13].

The main challenge in treatment is not inducing CR but maintaining it, as chemotherapy alone has proven ineffective in sustaining long-term response. Blinatumomab has shown efficacy in relapsed/refractory B-cell ALL and MRD-positive disease in CR, although it is not yet approved for first-line use. Blinatumomab has proven effective in maintaining MRD-negative status in older adults, achieving similar results to younger adults. The primary adverse events are cytokine release syndrome (CRS) and immune effector cell-associated neurotoxicity syndrome (ICANS), with these being more severe in older adults [14].

The SWOG study 1318, a phase II trial treating newly diagnosed Ph− B cell-ALL in adults over 65, used blinatumomab monotherapy for induction and POMP maintenance. An interim analysis at one year showed a CR rate of 66% and an EFS of 56%. Ongoing trials are exploring the combination of inotuzumab ozogamicin and blinatumomab in newly diagnosed and relapsed/refractory B-cell ALL in older adults [10].

Blinatumomab has also demonstrated benefits in sustaining CR in MRD-negative patients. A phase III trial involving adults with MRD levels below 0.01% after induction and intensification chemotherapy showed superior OS at 43 months (85% vs. 68%) for those receiving blinatumomab consolidation along with chemotherapy. The RFS was also superior (80% vs. 64%). However, neuropsychiatric symptoms were more common in the treatment group. As a result of these findings, blinatumomab was recently approved for use in adult patients with Ph− ALL who are in first CR with MRD-negative status, defined as MRD levels below 0.01% [15]. An algorithmic approach to the treatment of disease in medically fit patients is summarized in Figure 1.

## 3. Unfit Older Adults

For older adults deemed unfit for intensive therapy (typically aged over 75 years or with high comorbidity scores [CCI > 2, HCT-CI > 3]), treatment is largely palliative. Options include low-intensity regimens such as steroids, vincristine, and POMP chemotherapy, depending on patient tolerance [10].

Emerging non-chemotherapy regimens are being explored to improve outcomes in this population. One small study evaluated inotuzumab ozogamicin plus blinatumomab with a dexamethasone and vincristine backbone in elderly patients over 70 years, demonstrating a CR rate of 93% after one cycle, with 92% achieving MRD-negative status. Out of the 14 patients who were enrolled, 13 achieved a complete response after one cycle, and after two cycles of treatment, all 13 responders were MRD-negative by flow cytometry; 12 responders had MRD evaluated with NGS as well, and 11 of them were MRD negative at a sensitivity of 10^−6^. Moreover, at a median follow up of 15 months, the 1-year rate for RFS was 64%; for objective response (OR), it was 70%, and for OS, it was 73%. The median was not reached for any of these parameters at the time. Five patients had died, three of them in remission from pneumonia, MI, and respiratory failure (none of them from neutropenia). The other two died after relapse; one of them had a KMT2A mutation, and the other one had hypoploidy plus a TP53 mutation. There was one patient who developed grade 3 encephalopathy during treatment with blinatumomab that resolved with steroids, and blinatumomab was able to be reintroduced later on. There were no cases of VOD, but all patients were pre-treated with ursodeoxycolic acid. Despite this good adverse event profile, most patients did have grade 1–2 encephalopathy during treatment with blinatumomab [16].

There is another small case series report of nine elderly patients with relapsed/refractory pre-B cell ALL, including patients with Ph+ status, who received treatment with inotuzumab ozogamicin monotherapy; all patients included were CD22+ at the time of beginning treatment. Out of the nine patients, only six could be assessed for disease response (two were selected for CAR T cell harvesting, and one patient died of sepsis complications); of the six patients, five achieved complete response, three of whom achieved MRD-negative status [14]. Treatment of older adults who are not fit for intensive therapy is summarized in Figure 2.

## 4. Central Nervous System-Directed Therapy

Central nervous system (CNS)-directed therapy is crucial for managing adult ALL and preventing CNS relapse. When there are more than five white blood cells (WBCs) per microliter in cerebrospinal fluid (CSF) with the presence of blasts, it indicates CNS involvement. Historically, patients with traumatic lumbar puncture (LP) have been at higher risk for CNS relapse and poor event-free survival (EFS), as this procedure may theoretically “seed” the CNS with leukemia cells [10,17].

### 4.1. Central Nervous System Prophylaxis

For prophylaxis, intrathecal (IT) chemotherapy combined with intracranial radiation therapy (XRT) has been the standard of care for many years. This combination was proven in a phase III trial to reduce the rate of CNS relapse at two years to 19%, compared to 42% in the control group that did not receive CNS prophylaxis. However, the use of intracranial XRT has notable side effects, including seizures, cognitive deficits, and growth stunting. In adults, the combination of XRT with chemotherapy can lead to unacceptable myelotoxicity and other toxicities, making it difficult to proceed with timely consolidation therapy. As a result, the newer standard of care for most patients has shifted to a combination of IT chemotherapy and systemic chemotherapy, which includes high-dose methotrexate (MTX) and cytosine arabinoside, replacing XRT. Studies in adults show that this combination results in CNS relapse rates of approximately 5%. Although no single IT chemotherapy regimen is universally accepted, combinations of MTX, cytarabine, and glucocorticoids (Gc) have been used. While this combination may prolong CNS EFS, evidence suggests it does not significantly improve overall survival (OS). XRT may still be included in some protocols for high-risk patients, such as those with WBC counts greater than 100,000 or T-cell leukemia, although evidence from two meta-analyses does not support its use, as the risk for CNS relapse was not reduced in the XRT-treated groups [17].

### 4.2. Central Nervous System Treatment

When CNS leukemia is detected, treatment should include IT triple therapy (a combination of chemotherapy agents) 2–3 times a week until no blasts are detectable in the CSF. This treatment is typically combined with systemic therapy, which includes high-dose methotrexate, high-dose asparaginase, and dexamethasone. Several risk factors contribute to the likelihood of CNS leukemia, including mature B-cell and T-cell ALL, elevated lactate dehydrogenase (LDH) levels greater than 600, a high proliferative index at diagnosis (more than 14% of lymphoblasts in the S and G2/M phases), and high-risk cytogenetics such as KMT2A, NPM1, or Ph-positive ALL [10,17].

## 5. MRD-Based Treatment Modification

MRD-based prognostication is a critical component in the management of ALL and involves several key factors. The threshold for minimal residual disease (MRD) is typically set at <0.01%, as this aligns with the sensitivity of MRD detection methods. This threshold is important because it helps predict the time to hematological relapse. Patients with higher MRD levels are more likely to relapse sooner and may be candidates for MRD-based therapies. The timing of MRD testing also plays a significant role in its prognostic value. If MRD is tested during induction, a negative result is a good indicator of favorable outcomes and may even allow for treatment de-escalation. On the other hand, testing after the first consolidation is often considered the best time to assess the risk of relapse and to determine if a patient may benefit from a hematopoietic stem cell transplant (HSCT) [18].

The method of MRD measurement is another important consideration. Multiple PCR-based techniques can be used, as long as the laboratory has experience with methods like clonal immunoglobulin and T-cell receptor measurement. For MRD measurement to impact treatment decisions, it must have a minimum sensitivity of 0.01% detection. This sensitivity ensures that MRD findings are accurate enough to guide clinical choices [19,20].

Current uses of MRD detection include determining eligibility for stem cell transplantation (SCT). Patients with persistent MRD are generally considered good candidates for SCT, as it has been shown to offer a survival benefit. However, higher MRD levels are associated with an increased chance of relapse after SCT and more difficulty achieving complete remission (CR1) prior to transplantation. In general, patients with MRD levels greater than 0.1% after three cycles of standard therapy should be considered for SCT or targeted therapies. This approach has demonstrated a survival benefit, as previously indicated in various studies [21].

Early modification of treatment based on MRD findings is also crucial. Blinatumomab has proven to be effective in inducing remission and prolonging survival in adult ALL patients with MRD levels greater than 0.1%. A single-arm study conducted in 2018 demonstrated that patients in both first and later remissions could benefit from blinatumomab before proceeding to SCT. In this study, 78% of patients achieved complete molecular response (CMR) after just one cycle of blinatumomab. Achieving CMR was associated with significantly longer relapse-free survival (23.6 months vs. 5.7 months) and overall survival (OS) (38.9 months vs. 12.5 months). The median OS for all patients who had hematological response (including those with MRD -positive status) was 36.5 months, compared to only 6 months for those who did not have at least hematological response. Additionally, 67% of these patients went on to receive SCT, with a low relapse rate after the procedure. However, it is noteworthy that all patients who did not undergo SCT eventually relapsed, highlighting the importance of early treatment modification based on MRD results [20,21].

## 6. Hematopoietic Stem Cell Transplant in Adults

Hematopoietic stem cell transplantation (HSCT) remains the only curative option for adults with acute lymphoblastic leukemia (ALL), due to the high risk of eventual relapse. However, the evolving role of minimal residual disease (MRD) in predicting disease relapse is reshaping the indications for HSCT. While HSCT remains indicated in most cases with high-risk features, such as KMT2A rearrangements or t(4;11) translocations and Ph-positive disease, the criteria for transplantation have become more nuanced. Subtypes that were previously considered clear indications for HSCT, such as low hypoploidy, complex karyotypes, early T-cell precursor ALL (ETP-ALL), and Ph-like ALL, now present conflicting evidence regarding their survival benefit. The development of new treatment options, including blinatumomab and chimeric antigen receptor T-cell (CAR-T) therapies, has provided alternative strategies with less treatment-related mortality (TRM), challenging the historical reliance on HSCT [22,23,24].

MRD is increasingly being incorporated into decisions about HSCT. In younger adults, particularly those treated under adolescent and young adult (AYA) protocols, 5-year survival rates range from 60 to 70%, and these patients may be spared from HSCT if they achieve MRD negativity. On the other hand, patients considered to be at standard risk who have MRD positivity (greater than 0.01%) after three cycles of standard therapy should be considered for HSCT or targeted therapies, as this indicates a higher risk of relapse. Even patients who achieve MRD-negative status after maintenance therapy still face a 20–30% chance of hematological relapse, emphasizing the importance of frequent MRD monitoring in this patient population. However, the challenges of frequent MRD measurement include technical limitations and resource constraints, as well as the lack of a standardized MRD testing method [10,22].

HSCT remains a viable option for older adults, despite poor outcomes with non-transplant strategies. The primary challenge in this age group is treatment-related mortality (TRM). Reduced-intensity chemotherapy (RIC) regimens have shown similar overall survival (OS) outcomes to myeloablative regimens, offering the advantage of reduced TRM, although they may lead to higher relapse rates. A 2017 review by the Acute Leukemia Working Party (ALWP) of the European Bone Marrow Transplantation (EBMT) group evaluated the effects of HSCT at first complete remission (CR1) using RIC in older adults, regardless of Philadelphia chromosome (Ph) status. In this cohort, with a median age of 62 years, the 3-year OS was 42%, and event-free survival (EFS) was 35%. The most significant negative impact on OS was observed with cytomegalovirus (CMV) donor–recipient mismatch, and graft-versus-host disease (GVHD) was more common in patients with unrelated donors. These findings underscore the complexities of HSCT in older adults and the need for tailored treatment strategies to optimize outcomes [25].

## 7. Treatment of Relapsed Disease

The treatment of relapsed acute lymphoblastic leukemia (ALL) continues to be a challenging area, with allogeneic hematopoietic stem cell transplantation (HSCT) remaining the only curative option for patients who experience relapse. Relapses can occur either in the bone marrow or as extramedullary (EM) disease. EM disease relapse is particularly concerning, as it is associated with a worse prognosis. A clue to the presence of EM disease can be a positive minimal residual disease (MRD) result in the peripheral blood, even when the bone marrow is negative for MRD. For relapses that occur more than 18 months after the first complete remission (CR) (late relapses), the treatment approach typically involves repeating the initial induction regimen. This may include high-dose cytarabine (HiDAC) combined with anthracyclines or mithoxantrone, or FLAG-Ida, a regimen involving fludarabine, high-dose cytarabine, filgrastim, and idarubicin. High-dose methotrexate and high-dose cytarabine (Ara-C) are also options for these cases. For second or subsequent relapses, where the aim is not curative, liposomal vincristine is an approved treatment. Relapsed T-cell ALL requires a different approach, and nelarabine has shown promising results in this setting. It has demonstrated a 41% complete remission (CR) rate with a median CR duration of 3 months, although the overall survival (OS) remains limited, with 1-year survival at 28% and 3-year survival at 11% [26,27].

Immuno-oncology agents have emerged as a promising area of treatment for relapsed disease, although there is currently no standard of care (SOC) for their use, particularly due to overlapping indications. The evidence available for these therapies in first-line treatment, relapsed disease, and even MRD-negative disease is encouraging. However, these therapies are currently approved only for B-cell ALL. For T-cell ALL, the use of these agents is limited due to the risk of T-cell fratricide and immunosuppression, which makes their use unsuitable for this subtype [10,28].

### 7.1. Blinatumomab

As mentioned before, blinatumomab is approved for relapsed or refractory CD19-positive B-cell ALL. The TOWER trial was the first phase III trial to compare blinatumomab with chemotherapy in previously treated Ph-negative B-cell ALL. This trial showed a CR rate of 44% for blinatumomab compared to 25% for chemotherapy alone, with OS of 7 months versus 4.4 months. Factors that predicted poor response to blinatumomab included a low number of marrow blast cells, extramedullary disease, a high amount of circulating regulatory T cells (Tregs), and PD-L1 expression in B-cell blasts [14].

As previously stated, given the short duration of response with blinatumomab, it is recommended that patients undergo HSCT as soon as possible after achieving CR, with blinatumomab serving as bridging therapy [13,20].

### 7.2. Inotuzumab Ozogamicin

Inotuzumab ozogamicin (IO), which targets CD22 and delivers the chemotherapy payload calicheamicin, is approved for relapsed or refractory B-cell ALL in adults. The phase III INO-VATE trial demonstrated that IO resulted in a higher CR rate compared to chemotherapy (80.7% versus 29.4%) and a higher MRD-negative rate (78.4% versus 28.1%). In survival analyses, IO showed superior progression-free survival (PFS) (5.0 months versus 1.8 months) and marginally superior OS. Notably, more patients in the IO group underwent allo-HSCT, and this subgroup experienced significant leukemia-free survival. However, a major concern with IO is VOD, which complicates the assessment of hepatotoxicity, especially in patients who undergo HSCT. Overall, serious adverse events and hematological side effects were less frequent with IO compared to chemotherapy. A post hoc analysis comparing younger adults (<55 years) with older adults (>55 years) in the INO-VATE study showed similar CR rates (70% versus 75%) and MRD negativity (79% versus 76%) across age groups, with no significant difference in the incidence of SOS. These findings highlight the potential for IO as a treatment option in both younger and older adult populations with relapsed B-cell ALL [13].

### 7.3. Chimeric Antigen Receptor T Cells (CAR-T)

Chimeric antigen receptor T-cell (CAR-T) therapy has emerged as an exciting and promising treatment for relapsed or refractory (RR) acute lymphoblastic leukemia (ALL). Initial approvals for CAR-T targeting CD19 were based on the ELIANA trial, which demonstrated efficacy in children and young adults. However, the safety and toxicity profile for adults remains under further investigation. A phase II study is being prepared to evaluate the use of CAR-T therapy in adults, with revisions to the toxicity profile underway [29].

The ZUMA-3 trial, released in 2021, focused on adults with RR B-cell ALL, including those who had relapsed after stem cell transplantation (SCT). This multicenter phase II trial involved 71 patients, with 55 patients successfully receiving the treatment. The median age of the treated patients was 40 years, with 15% of patients over the age of 65. The results from this trial showed that 56% of treated patients achieved complete remission (CR), and 71% achieved CR with incomplete hematological recovery. The median duration of response was 12.8 months, with a median relapse-free survival (RFS) of 11.2 months. Despite these promising results, 20 patients died, mostly due to disease progression, with two deaths linked to grade 5 toxicities, including brain herniation and septic shock. A significant proportion of patients (approximately 24%) experienced grade 3 cytokine release syndrome (CRS) or immune effector cell-associated neurotoxicity syndrome (ICANS), highlighting the need for ongoing monitoring of these side effects [25,30].

In 2023, long-term follow-up data from the ELIANA trial indicated a decline in the likelihood of maintaining RFS over time. While 81% of patients maintained CR at the 1-year mark, only 51% maintained RFS. The main cause of relapse was CAR-T cell depletion or the persistence of CD19 expression, which is why loss of CD19 expression on B-cells remains a significant concern. To address this, new developments in multi-target CAR-T cells are underway, including constructs that co-target CD19 and CD22. One of the largest studies to date on these dual-target CAR-T cells is a phase 1 trial for CD22 CAR-T cells in RR B-cell ALL patients who have relapsed after CD19-targeted CAR-T therapy [28]. Early results show manageable toxicity and promising outcomes, although it remains too early to draw definitive conclusions. It is worth noting that targeting CD22 with CAR-T cells carries an increased risk of hemophagocytic lymphohistiocytosis (HLH) compared to CD19 therapy, though it appears to reduce the incidence of neurotoxicity and ICANS [29,31,32].

### 7.4. Immuno-Oncology Agents for Extramedullary and CNS Disease

When considering the use of immuno-oncology agents for extramedullary (EM) or central nervous system (CNS) disease, it is important to note that antibody-based therapies like blinatumomab or inotuzumab ozogamicin (IO) have poor blood–brain barrier (BBB) penetration, limiting their effectiveness for these disease sites. These therapies must therefore be combined with intrathecal (IT) chemotherapy to address CNS or EM disease. On the other hand, CAR-T cells exhibit good CNS penetrance, making them a potentially effective option for patients with CNS disease burden. However, patients with a high CNS disease burden are at increased risk for neurotoxicity, which may negate the survival benefits of CAR-T therapy. The experience with CAR-T cells in CNS disease is still limited, with small studies and short follow-up periods. A meta-analysis supports the findings that CAR-T cells can effectively treat CNS involvement, but further research and data are needed to confirm their long-term efficacy and safety in this setting [14,33].

### 7.5. Miscellaneous Alternatives for Relapsed/Refractory Disease

These are therapeutic approaches that have been used seldom for the management of ALL, but with promising current data.

#### 7.5.1. B-Cell Lymphoma 2 (BCL-2) Homology 3 (BH3) Mimetics 

BH3 mimetics are a class of drugs that bind to and inhibit anti-apoptotic BCL-2 proteins. These drugs engender leukemic cell death through permeabilization of the mitochondrial outer membrane. One of the most commonly employed BH3 mimetics is venetoclax, typically used in the treatment of acute myeloid leukemia and chronic lymphocytic leukemia. Preclinical studies have demonstrated the efficacy of various BH3 mimetics, such as venetoclax and navitoclax, in B-lineage ALL cells [8]. Subsequent clinical trials have investigated the use of venetoclax in the treatment of ALL, primarily in relapsed or refractory cases. One phase 1/2 study looked at the combination of venetoclax with mini HCVD in patients with relapsed/refractory disease, which resulted in an overall survival of 7.1 months and a 1-year overall survival rate of 29% [34]. Another recent study also evaluated venetoclax with mini-HCVD, both in relapsed/refractory patients as well as newly diagnosed patients with ALL. Complete remission was achieved in 90.9% of patients with newly diagnosed disease and 37.5% of patients with relapsed/refractory disease [35]. These studies are encouraging, and further development of venetoclax-based combinations in ALL is warranted. Other BH3 mimetics, such as navitoclax, are being actively studied in ALL as well [36].

#### 7.5.2. Daratumumab

Daratumumab is an anti-CD38 monoclonal antibody primarily used in the treatment of multiple myeloma; however, there have been some recent studies exploring its use in ALL, both of B-cell and T-cell origin. The DELPHINUS study investigated the use of daratumumab in relapsed/refractory B-cell and T-cell pediatric ALL. Results were promising for the cohort of patients with relapsed/refractory T-cell ALL, with complete remission rates of 41.7% in childhood T-cell ALL and 60% in young adult T-cell ALL. The efficacy of daratumumab in B-cell ALL remains limited [37]. Robust data regarding the use of daratumumab in older adults with ALL have yet to be obtained. Some of the challenges are that ALL in older adults is a relatively uncommon illness, and there is a widely heterogenous range of options that are attempted in cases of relapsing/refractory ALL in adults, but the studies that are looking at these patients are usually small and focus on younger adults. Moreover, there is a significant potential for positive-outcome bias, given the unrelenting nature of ALL once it has not responded to several lines of treatment, and efforts are made to publish positive results when available [38].

## 8. Ph+ ALL

Philadelphia chromosome-positive acute lymphoblastic leukemia (Ph+ ALL) is a genetically distinct and aggressive leukemia subtype with the presence of the BCR-ABL fusion gene. In this section, we will synthesize advancements in its treatment, focusing on the transformative role of tyrosine kinase inhibitors (TKIs), the integration of immunotherapy, and evolving strategies for adult populations [39].

The introduction of TKIs has revolutionized the treatment of Ph+ ALL, targeting the BCR-ABL fusion protein’s tyrosine kinase activity. TKIs, such as imatinib, dasatinib, and ponatinib, have significantly improved outcomes by achieving remissions with reduced toxicity compared to traditional chemotherapy. Imatinib, when combined with low-intensity chemotherapy, demonstrated high remission rates and is the primary treatment for younger patients. Ponatinib, noted for its efficacy against T315I mutation, remains a critical option for resistant cases or relapse. The combination of TKIs with immunotherapies, such as blinatumomab (a bispecific T-cell engager targeting CD19), is now being explored to reduce reliance on intensive chemotherapy and stem cell transplantation (SCT) [39,40]. Recent advances suggest that integrating TKIs with immunotherapy or reduced-intensity chemotherapy achieves comparable outcomes to traditional regimens. For example, studies highlight the success of chemo-free regimens that pair TKIs with agents like blinatumomab, providing a pathway for treatment in older patients unfit for intensive therapies. Such approaches reduce treatment-related toxicities while maintaining excellent survival rates [40,41].

Allo-HSCT has been considered the standard for curing Ph+ ALL. However, the necessity of transplantation in the first complete remission is increasingly questioned with TKIs and immunotherapies. Currently, research undermines the minimal residual disease (MRD) monitoring to guide decisions about transplant eligibility. Patients may forgo allo-HSCT, especially when treated with next-generation inhibitors like ponatinib or antibody-based therapies such as inotuzumab ozogamicin [39,41].

Immunotherapies, including blinatumomab and inotuzumab ozogamicin, are central to evolving Ph+ ALL management. These agents enable chemo-free regimens, particularly in patients unfit for transplantation or those with relapsed disease. Chimeric antigen receptor (CAR) T-cell therapies targeting CD19 are under investigation, offering a promising option. Furthermore, next-generation precision medicine is paving the way for personalized treatment approaches by identifying resistance mutations and tailoring therapies [41,42].

*Ph*+ ALL with p190 mutation, for example, is a distinct subtype of ALL characterized by the presence of the *BCR-ABL1* fusion gene and the production of tyrosine kinase p190 BCR-ABL1 oncoprotein. Other *BCR-ABL1* fusion chimeric proteins include p210 and p230. The mutation is typically associated with a more aggressive disease course and is a poor prognostic marker; however, the advent of TKIs has significantly improved outcomes for this subtype. The poor prognosis historically associated with *Ph+* ALL has dramatically shifted with the use of TKIs, as response rates to first-line therapies are now more closely resembling *Ph−* disease [43]. Unfortunately, *IKZF1* deletions are present in over 80% of *Ph+* ALL cases and are associated with resistance to TKIs and poor prognosis. Particularly, the presence of *IKZF1^plus^*, defined as a *IKZF1* deletion with concurrent deletion of additional genes such as *PAX5*, is associated with an even higher risk of treatment failure and relapse. Concurrent, more-aggressive therapy with a second or third generation TKI and blinatumomab should be considered if *IKZF1^plus^* is identified [44,45].

The development of TKIs, immunotherapy, and precision medicine has significantly changed the treatment landscape for Ph+ ALL. While TKI-based approaches and chemo-free regimens are changing the standards of care, allo-HSCT is still an option for high-risk or relapsed cases. Future studies should keep improving these strategies to guarantee the best results with the least amount of treatment-related side effects for all types of patients. Treatment of Ph+ ALL is summarized in Figure 3.

## 9. Ph-like ALL

One subset of patients not included above is patients afflicted with *Philadelphia chromosome-like (Ph-like)* ALL. Patients with the disease are denoted as such given a genetic expression profile resembling *Ph+* ALL; however, no *BCR-ABL1* fusion gene is observed in this high-risk subtype defined by genetic changes leading to aberrant kinase signaling [46].

Instead, *Ph-like* ALL is characterized by alternative genetic mutations. One subclass, for example, includes ABL-class fusions, involving genes such as *CSF1R*, *PDGFRB*, *ABL1*, and *ABL2*. The aforementioned fusions often mimic the typical *BCR-ABL1* fusion noted in *Ph*+ ALL. Other common mutations include alterations in *CLFR2* alterations or the JAK-STAT pathway [47].

Patients with *Ph-like* ALL have a significantly worse prognosis, with various studies demonstrating a 5-year overall survival of approximately 23% [48]. The worse overall and event-free survival can be attributed to persistent MRD and higher rates of induction failure and relapse [49].

In regard to treatment, similar to the above, a key emerging area of focus is on personalized molecular targeting agents. Next-generation sequencing can uncover specific driver mutations that are amenable to therapy, such as the use of TKIs in patients exhibiting *ABL*-class mutations. TKIs are pivotal in treating *Ph-like* ALL by targeting dysregulated signaling pathways. For example, dasatinib and ruxolitinib have shown efficacy in managing BCR-ABL1-like and JAK/STAT pathway abnormalities, respectively. Their integration into combination regimens with traditional chemotherapy significantly improves outcomes, as highlighted in studies focusing on pediatric and young adult populations [50,51].

Furthermore, data suggest that TKIs may address persistent minimal residual disease (MRD), a frequent complication associated with poor survival in this subtype [48,49].

Given that the advent of such molecular testing is relatively new, a standard-of-care approach to the treatment of Ph-like ALL has yet to be delineated. JAK inhibitors have also shown promise in preclinical studies and case series for patients with *CRLF2* muta-tions and JAK-STAT pathway alterations, and multiple clinical trials in this area are ongoing. The combination of these targeted therapies with chemotherapy is another area of ongoing research and promise [50,52].

Emerging immunotherapies, including monoclonal antibodies and CAR-T cell therapies, offer promising options for refractory or relapsed cases, such as blinatumomab, which is a bispecific T-cell engager targeting CD19 and has demonstrated efficacy in inducing remission among relapsed *Ph-like* ALL cases [7,46]. Inotuzumab ozogamicin, an antibody–drug conjugate that targets CD22, has been proven to be effective in cases unresponsive to conventional therapies [7,50]. While they are still under investigation, CD19-directed CAR-T therapies show potential in overcoming resistance, particularly for patients with relapsed or refractory disease [7,46]. CAR T-cell therapy has demonstrated remission rates up to 90% in heavily pretreated patients with Ph-like disease.

Despite advances, treatment outcomes for *Ph-like* ALL remain inferior compared to other ALL subtypes. Persistent MRD, noted in trials such as GIMEMA LAL1913, highlights the need for intensified or novel therapies [49]. Additionally, the genetic heterogeneity of *Ph-like* ALL complicates standardized treatment. Comprehensive molecular profiling is essential for tailoring therapy to specific alterations, such as CRLF2 rearrangements or *ABL*-class fusions [48,51].

Some things can be achieved to overcome these challenges. Enhanced molecular diagnostics would allow for broader access to advanced diagnostic tools, which will enable earlier detection and personalized treatment [46,48]. Further development of TKIs and monoclonal antibodies is critical to expanding treatment options for diverse genetic subtypes [50]. Researching more on CAR-T cells and bispecific antibodies may redefine outcomes for relapsed or refractory *Ph-like* ALL [7,36]. Finally, figuring out solutions to some critical global disparities can ensure the availability of cutting-edge therapies in environments that have limited resources [48,49].

Overall, while significant progress has been made in treating *Ph-like* ALL, particularly through TKIs and immunotherapies, challenges such as MRD persistence and genetic diversity remain. Continued research and innovation are essential to improve survival and quality of life for patients with this high-risk leukemia subtype.

## 10. Conclusions

As the prognosis for ALL in adults remains unsatisfactory when compared to pediatric populations, novel therapeutics targeting specific biomarkers are being increasingly used in the relapsed or refractory setting. Using these therapies in the frontline setting is an emerging area of much promise. In particular, frontline integration of immuno-oncology agents such as bispecific T-cell engagers and antibody–drug conjugates is an exciting prospect in regard to maintaining or improving outcomes and efficacy while also minimizing toxicity; future research should be directed towards this area.

## Figures and Tables

**Figure 1 cells-14-00371-f001:**
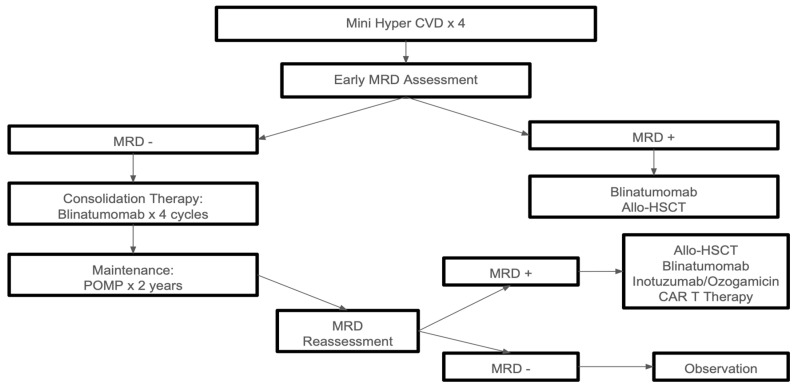
Treatment approach for medically fit adults with *Philadelphia chromosome negative (Ph−)* ALL. Legend: Hyper-CVAD (chemotherapy regimen); MRD (minimal residual disease); POMP (chemotherapy regimen); Allo-HSCT (allogeneic hematopoietic stem cell transplant); CAR T (chimeric antigen receptor T-cell).

**Figure 2 cells-14-00371-f002:**
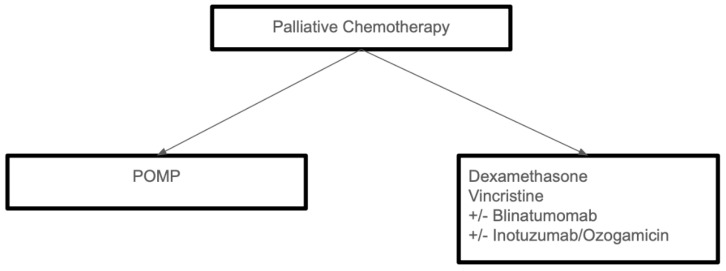
Treatment approach for medically unfit adults with *Ph−* ALL. Legend: POMP (chemotherapy regimen).

**Figure 3 cells-14-00371-f003:**
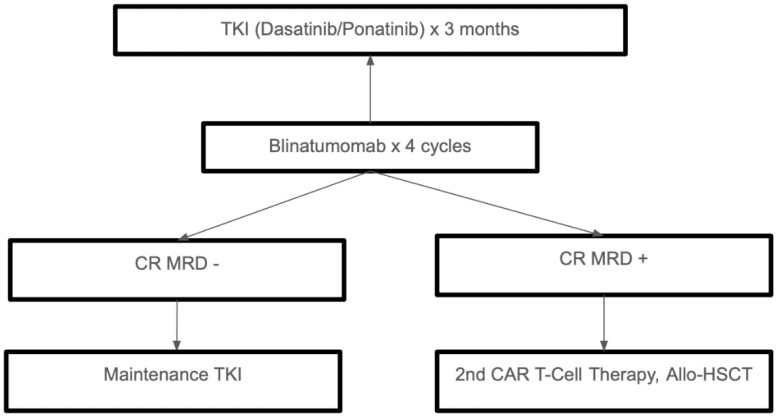
Treatment approach in adults with *Philadelphia chromosome positive (Ph+)* ALL. Legend: TKI (tyrosine kinase inhibitor); CR (complete remission); MRD (minimal Residual disease); Allo-HSCT (allogeneic hematopoietic stem cell transplant); CAR T (chimeric antigen receptor T-cell).

## Data Availability

No new data were created or analyzed in this study.

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
