# Peer review of "Advances in Therapy of Adult Patients with Acute Lymphoblastic Leukemia"

_cells, 2025, doi:10.3390/cells14050371_

Round 1
Reviewer 1 Report
Comments and Suggestions for Authors
“Advances in Therapy of Adult Patients with Acute Lymphoblastic Leukemia” by Oscar Sucre is an interesting paper outlining many therapeutic approaches in this disease that afflicts patients frequently experiencing disease relapse. The statement in the abstract “This review seeks to outline and expand the current standard of care, as well as new advances, in the treatment of adult patients with ALL, and addressing future areas of research” is very promising, but some aspects are to be addressed.
-Other therapies, such as BH3 mimetics, are in ongoing clinical trials in adult ALL, and this aspect should be considered and discussed in the manuscript.
-We are conscious that immunotherapy is of undoubted validity in future cures.
Other monoclonal antibodies, such as daratumumab, should be discussed in the manuscript for a more comprehensive review.
- The manuscript would benefit from better analyzing future research areas to develop novel therapeutic strategies to target ALL.
Minor comments:
Lines 350-352 page 8. After the sentence: “One of the largest studies to date on these dual-target CAR-T cells is a phase 1 trial for CD22 CAR-T cells in RR B-cell ALL patients who have relapsed after CD19-targeted CAR-T therapy.” a reference is required and would help the reader.
Reviewer 2 Report
Comments and Suggestions for Authors
Sucre and c in "Advances in Therapy of Adult Patients with Acute Lymphoblastic Leukemia" well show the therapeutic approach with a clear algorythm. However, the work needs more clarity on some points 1) well define the population of fit older patients and unfit older patients. What is the used score, CI, Sorror etc, which age/range 2)advanced therapy means to pay attention on immunotherapy. The manuscript needs to be focused on immunotherapy. Blinatumumab has a crucial role on ALL treatment. Also inotuzumab should be better insert in the therapeutic algorythm, non only mentioned 3)the delicate role of asparaginase, in advanced therapy for ALL should be better discussed 4)for ALL Ph+ is important define the state of the art in this era: the indication to allogeneic transplant is always recommended, there is no mention in the paper about the mutation status, like IKarus plus /PAX5. The reader can understand that the allogeneic transplant is now an old option for adult ALL Ph+ 5)the role of CART in therapeutic algorythm in ALL: is now clearer, like a bridge to allogeneic transplant in particular setting 6) about the subdivision into paragraphs, I would prefer to read a longer introduction about the state of the art in young adult ALL field, then to focus on CNS prophylaxis and treatment, then focus on older popuilation and eventually to discuss about the new options on ALL therapy, with a special focus on immunotherapy and CART/allogeneic transplant. There are a lot of review on ALL treatment, the reader do not need the same story but something new about "advanced therapy" Good job!
The review coul be interesting but in this format is not attractive
The reader needs something new, expecially when in the title highlights "advanced therapy"
there are a huge of publications about use of immunotherapy in ALL, the reader would know something new about the role of CART/allogeneic transplants.
Also to focus role of asparaginase in ALL could be more attractive
I asked major revision, I'm sure that it could became a valuable paper
Round 2
Reviewer 1 Report
Comments and Suggestions for Authors
The Authors replied to all the concerns.